# Long-Read Transcriptome of Equine Bronchoalveolar Cells

**DOI:** 10.3390/genes13101722

**Published:** 2022-09-25

**Authors:** Sophie Elena Sage, Pamela Nicholson, Tosso Leeb, Vinzenz Gerber, Vidhya Jagannathan

**Affiliations:** 1Swiss Institute of Equine Medicine, Department of Clinical Veterinary Medicine, Vetsuisse Faculty, University of Bern, 3001 Bern, Switzerland; 2Next Generation Sequencing Platform, University of Bern, 3001 Bern, Switzerland; 3Institute of Genetics, Vetsuisse Faculty, University of Bern, 3001 Bern, Switzerland

**Keywords:** Iso-Seq, lung, bioinformatics, annotation, asthma, horse, *Equus caballus*

## Abstract

We used Pacific Biosciences long-read isoform sequencing to generate full-length transcript sequences in equine bronchoalveolar lavage fluid (BALF) cells. Our dataset consisted of 313,563 HiFi reads comprising 805 Mb of polished sequence information. The resulting equine BALF transcriptome consisted of 14,234 full-length transcript isoforms originating from 7017 unique genes. These genes consisted of 6880 previously annotated genes and 137 novel genes. We identified 3428 novel transcripts in addition to 10,806 previously known transcripts. These included transcripts absent from existing genome annotations, transcripts mapping to putative novel (unannotated) genes and fusion transcripts incorporating exons from multiple genes. We provide transcript-level data for equine BALF cells as a resource to the scientific community.

## 1. Introduction

Equine asthma is a naturally occurring disease of the horse with many similarities to human asthma. The etiology and pathophysiology of equine and human asthma are not completely understood. Bronchoalveolar lavage fluid (BALF) cytology is typically used to confirm the diagnosis in horses [1]. BALF cells are often used to study gene expression in models of asthma. Compared to mice or humans, the collection of BALF in horses is less invasive, as it can be performed in a standing animal with light sedation. This makes horses a valuable animal model for the discovery of novel pathways implicated in asthmatic inflammation [2,3,4]. To take full advantage of the comparative gene expression data gained from the equine asthma model, it is critical to have an accurate and comprehensive annotation of the genes expressed in equine BALF cells.

Short-read RNA sequencing (RNA-seq) is commonly used to study the changes in gene expression associated with a disease (e.g., equine asthma) [5,6,7]. This method allows for quantification of transcript abundance in a biological sample at the gene level, and sometimes even at the isoform level. However, short-read data are not sufficient to infer accurate full-length transcript structures, as transcripts with multiple alternative splicing sites cannot be completely covered by the individual short reads. Complete cDNA molecules can now be sequenced using long-read sequencing platforms such as Pacific Biosciences [8] and Oxford Nanopore [9]. In contrast to short-read sequencing, long-read-based techniques allow unambiguous mapping to a reference assembly, conservation of isoform information, as well as discovery of novel genes, fusion transcripts and non-coding RNA. This has led to significant improvement of the transcriptome annotation in a few domestic species such as chicken [10], rabbits [11] or pigs [12]. The FAANG consortium, whose goal is to create reference functional maps of livestock animal genomes [13] is already taking advantage of long-read sequencing. Recently, Peng et al. [14] applied the PacBio Iso-Seq sequencing technique to nine different horse tissues, including lung. They combined their Iso-Seq transcriptome with the existing NCBI and Ensembl EquCab3.0 reference annotations. The resulting FAANG equine transcriptome comprises 36,239 protein-coding genes and 153,492 transcripts, a considerable improvement compared to the current NCBI annotation, which contains only 21,129 protein-coding genes and 72,102 transcripts. The discovery of novel genes and isoforms with potential biological relevance will benefit to all genomics studies.

Advanced transcriptomics are becoming increasingly popular to study the pathophysiology of equine asthma. Our group recently performed the first single-cell mRNA sequencing of equine BALF [15], a technique that is widely used to study the transcriptome of individual cells and compare them. Peng et al. [14] have also showed that major isoform (based on relative expression values) differs between tissue type, hence a BALF-specific reference transcriptome is highly desirable.

In this study, we applied the Pacific Biosciences Iso-Seq method [8] to generate a full-length transcriptome of BALF cells originating from a healthy and an asthmatic horse. The dataset obtained is a valuable resource for future transcriptomic studies in horses, including investigations of equine asthma.

## 2. Materials and Methods

### 2.1. Study Animals

Privately owned horses were prospectively enrolled for a study on equine asthma. Suitable candidates were identified through a validated owner questionnaire (HOARSI) [16]. Requirements for study enrollment were an age ≥ 5 years old, a longer than two-month history of being fed hay, no history of immunotherapy, no history of upper airway disease (e.g., roarer), no evidence of current systemic disease and a rectal temperature ≤ 38.5 °C on the day of the exam. Additionally, the horses should not have received any corticosteroids, bronchodilators or anti-histaminic administration nor suffered from a respiratory infection in the two weeks preceding the examination. Owners were asked to bring a healthy horse (without respiratory signs) from the same barn along with their asthmatic horse to the Bern ISME equine clinic. Horses underwent the following standard diagnostic procedures to characterize their respiratory status: physical examination with clinical scoring [17], lower airway endoscopy with tracheal mucus scoring [18], and bronchoalveolar lavage. Selection of the samples for RNA extraction was based on the BALF quality (yield ≥ 30%, foamy, normal CASY cell counter profile (OMNI Life Science, Bremen, Germany)) and on the horses’ historical, clinical and laboratory features, with the goal of selecting the most archetypal phenotypes for both control and severe equine asthma (sEA). RNA was extracted from four BALF samples (two control and two sEA). One control and one SEA sample were then selected for sequencing based on the quality of RNA (see paragraph 2.5). The main characteristics of the sequenced horses are shown in Table 1. Detailed clinical features of these horses can be found in Appendix A.

### 2.2. Sample Collection

Horses were sedated with detomidine 0.01 mg/kg IV (Equisedan ^®^, Graeub, Bern, Switzerland) and butorphanol 0.01–0.02 mg/kg IV (Morphasol-10 ^®^, Graeub, Bern, Switzerland). A sterile BALF tube (300 cm; 10 mm outer diameter; Bivona Medical Technologies, Gary, Indiana, USA) was passed into the nose down to the trachea. Twenty ml of lidocaine 2% were instilled and the tube was further inserted until wedged against a peripheral bronchus. The cuff was insufflated with 5 mL of air and 0.9% NaCl (250 mL) was infused using 60-mL syringes. The fluid was re-aspirated until no more was yielded, at which point the cuff was deflated and the tube pulled out. The syringes content was pooled in a cooled silicone-coated glass bottle. The BALF was filtered through a 40-µm Falcon cell strainer and kept on ice until processing.

### 2.3. Sample Cryopreservation and Thawing

The protocol used to freeze and subsequently thaw the BALF cells has been described in detail in a previous study from our group [15]. The four samples from which RNA was extracted were stored at −80 °C for a median of 321 days (interquartile range [IQR] 50) before thawing.

### 2.4. RNA extraction and cDNA Library Preparation

Total RNA was extracted with the RNeasy mini kit (Qiagen, Hilden, Germany) according to the manufacturer’s instructions. The quantity and quality of the isolated RNA was assessed using a Thermo Fisher Scientific Qubit 4.0 fluorometer with the Qubit RNA BR Assay Kit (Q10211, Thermo Fisher Scientific, Reinach, Switzerland) and an Advanced Analytical Fragment Analyzer System with a Fragment Analyzer RNA Kit (DNF-471, Agilent, Basel, Switzerland). The RNA was also tested by spectrophotometry using a Denovix DS-11 FX (M/F) spectrophotometer/fluorometer to assess the purity of the RNA (DeNovix, Wilmington, DE, USA). The median RIN for the four samples was 9.85 (IQR 0.13). The two highest quality samples (RIN ≥ 9.9), representing one asthmatic and one control horse, were equally pooled based on their RNA concentration.

### 2.5. PacBio Iso-Seq Long-Read Sequencing

Once all quality control tests confirmed high quality of the pooled RNA sample, the Procedure & Checklist—Iso-Seq Express Template Preparation for Sequel and Sequel II Systems was followed (PN 101-763-800 Version 02 (October 2019)) using the standard protocol (Pacific Biosciences, Menlo Park, CA, USA). In short, full-length cDNA from 300 ng of total RNA was prepared using the NEBNext Single Cell/Low Input cDNA Synthesis & Amplification Module Kit. Thereafter, a SMRTbell express template prep kit 2.0 was used to prepare the library. The resulting library was Single Molecule Real-Time (SMRT) sequenced using a Sequel binding plate 3.0, sequel sequencing plate 3.0 with a 20 h movie time on a PacBio Sequel system using a SMRT cell 1M v3 LR. The 2.3 kb library was loaded at 4.5 pM and generated 28.6 Gb and 419,674 polymerase reads. Next, the Circular Consensus Sequencing (CCS) application was run in SMRT Link v10.1 (Pacific Bioscience, Menlo Park, CA, USA) using the continuous long reads sub-read dataset and default parameters. This resulted in 313,563 HiFi reads and 805,344,539 bp of HiFi yield. The HiFi mean read length was 2568 bp and 19 HiFi passes was recoded (mean). These HiFi reads were used to run the Iso-Seq analysis pipeline in SMRT Link v10.1. This generated 317,797 Full-Length Non-Concatamer (FLNC) reads with 5’ and 3’ primers and poly-A tails. The mean length of the FLNC reads was 2536 bp. Reads identified as full-length non-chimeric (FLNC) were considered for de novo clustering to generate unique isoforms. All steps, listed above, including RNA quality control assessments, were conducted at the Next Generation Sequencing Platform, University of Bern.

### 2.6. Isoform Sequence Analysis

Unique high-quality isoforms (supported by at least two FLNC) were mapped to the equine genome (version EquCab3.0) using minimap2 [19]. The following parameters were used with minimap2: (-ax splice -t 30 -uf --secondary=no -C5). The mapped data in SAM format were annotated using cDNA_Cupcake (https://github.com/Magdoll/cDNA_Cupcake;accessed on 14 February 2022 ) and SQANTI3 [20] (https://github.com/ConesaLab/SQANTI3; accessed on 14 February 2022) pipelines. The mapped data in SAM format were input to Cupcake in order to collapse redundant isoforms into transcript loci. 5′-differences were not considered when collapsing the isoforms. 5′-degraded isoforms were removed using the filter_away_subset function of cDNA_cupcake. The resulting isoforms were annotated and compared with EquCab3.0 NCBI annotation release 103 using SQANTI3 with default parameters. The known and novel isoforms were categorized by SQANTI3 into nine different structural classes (see Table 2). A schematic overview of the bioinformatics workflow is shown in Figure 1.

Template switching during reverse transcription of cDNA is a known source of false non-canonical splice junctions [21,22]. Secondary structures in RNA template or direct repeats can cause reverse transcriptase (RT) to switch from one template to the other, creating gaps in the cDNA sequence. The gaps are interpreted as splicing-like events by algorithms, thereby artificially inflating for non-canonical splice sites. To detect this RT switching event, SQANTI3 looks for repeat sequences flanking the non-canonical splicing exon-intron boundary. The output of SQANTI3 classification was filtered for transcripts that had 1) a junction classified as a RT-template switching artifact and 2) 86% or higher adenosine content in the 20 nucleotides immediately downstream of the aligned 3′-end of the transcript, as well as a continuous run of eight or more A (i.e poly(A)), indicating a possible oligo(dT) intra-priming artifact. These transcripts were filtered using SQANTI3 filtering script sqanti3_Rulesfilter.py.

Saturation-discovery or rarefaction curves were produced by subsampling full-length reads at different depths. Full-length reads were randomly sampled and, for each subsample of reads, the number of unique genes or transcript isoforms detected was determined. For each sampling depth, 100 sampling iterations were performed before computing the average number of unique genes or isoforms observed. Only isoforms exactly matching the NCBI annotation release 103 were retained. The saturation-discovery curve analysis was produced with the ‘subsample.py‘ and ‘subsample_with_category.py‘ scripts available in the cDNA_cupcake repository.

### 2.7. BUSCO Analysis

Benchmarking Universal Single-Copy Orthologs (BUSCO) [23] looks for near-universal single-copy orthologs present in a whole transcriptome dataset. We used BUSCO to determine the percentage of orthologs present in equine BALF. BUSCO v4.1 was run with its default settings, using the BUSCO vertebrate database (https://busco.ezlab.org/; accessed on 8 June 2022).

### 2.8. Blastp Analysis

SQANTI3 uses the GeneMarkS-T (GMST) algorithm to predict ORFs from the transcripts. The predicted ORFs were mapped to the clustered nr protein database [24] using blastp with the parameters -max_target_seqs 5 -evalue 0.0001 -outfmt “6 qseqid qaccver qlen sseqid saccver slen length pident qcovs evalue bitscore stitle”. The ORFs were also mapped to the PFAM database using the HMMER algorithm [25]. The predicted fusion transcripts were searched against the ConjoinG database [26]

### 2.9. Gene ontology (GO) Analysis

EnrichR online tool [27]-based gene enrichment analysis was performed with the top 500 most abundantly expressed genes. Total number of full-length read counts associated with genes was used as expression value. This value was obtained from the SQANTI3 classification file. The functional categories examined were: GO_Biological_Process 2021, GO_Cellular_Component 2021, GO_Molecular_Function 2021, Jensen TISSUES and PanglaoDB Augmented. Jensen tissues is a human database of gene–tissue associations. It records the expression of mRNA or corresponding protein in several tissues collected from multiple sources using various data types. PanglaoDB is a single cell RNA-seq database for mouse and human.

## 3. Results

### 3.1. Full-Length Transcripts

PacBio Iso-Seq data were generated from pooled RNA isolated from BALF cells of one healthy and one severely asthmatic horse (See Table 1 and Appendix A for details). We collected 12,487,670 subreads comprising ~ 51 Gb of raw sequencing data to generate consensus reads with the circular consensus sequencing (CCS) technology (Figure 1A). The CCS dataset consisted of 313,563 Hi-Fi reads with a mean length of 2568 bp (Figure 1B and Table 3) and a median quality of Q37. Adapter and poly(A) tail removal resulted in 258,022 FLNC sequences. Clustering and polishing of FLNC sequences produced 20,462 high quality (HQ) sequences and 159 low quality consensus transcript sequences. The HQ sequences had an accuracy > 99% and were supported by at least 2 FLNC reads. The HQ sequences had a mean length of 2931 bp.

Rarefaction analysis with subsampled full-length reads showed that sequencing saturation for transcript loci discovery was almost attained. Therefore, a higher sequencing depth would have been unlikely to detect more transcripts (Appendix A).

### 3.2. Annotation of HQ Isoforms

All HQ isoforms were mapped to the EquCab3.0 reference genome. The resulting SAM file from the transcript mapping was used for collapsing redundant transcripts isoforms based on genomic location. The collapsed data showed 17,538 isoforms mapping to unique locations and 599 HQ transcripts that did not map, had low alignment length coverage (<99%) or had low alignment identity (<95%). Each of the 17,538 unique isoforms was supported by at least two full-length reads. Filtering of the 1484 isoforms with truncated 5’-ends left 16,054 isoforms for the next analysis steps.

### 3.3. Isoform Characterization

The Iso-Seq isoforms aligned to 58.5% (n = 1960) vertebrate orthologs, of which 36.1% (n = 1210) matched single-copy orthologs.

Transcripts with unreliable 3′-ends (intra-priming) and/or with junctions labeled as (i) RT-switch, (ii) without a minimum coverage of three reads were filtered (n = 1820). The numbers of filtered transcripts from each structural category are shown in Appendix A.

The final filtered Iso-Seq dataset consisted of 14,234 transcript isoforms covering 6880 previously annotated and 137 novel genes (Table 4). Of those, 6309 (90.0%) were protein-coding genes. Most of the transcripts identified were protein-coding (98.0%).

Majority of transcripts were FSM isoforms (66.5%), while ISM isoforms represented 9.4% of the transcripts. NIC and NNIC accounted for 13.2% and 9.4% of the transcripts, respectively (Table 5). The 35 isoforms of the *CD163* gene transcript are depicted in Figure 2A as an example. They consisted of four FSM, two ISM, 12 NIC and 17 NNC isoforms. Of the NIC and NNC isoforms, 20 showed intron retention.

The full-length transcripts recovered had a length up to 10 kb (Figure 1B). The longest transcript was 10,514 bp in length and mapped to the gene *ABCA1*, which encodes a protein that exhibits ATPase-coupled transmembrane transporter activity. A total of 3372 (49%) genes were multi-exonic and gave rise to more than one isoform. A total of 163 (2.4%) of the 6880 known genes identified had more than six isoforms. The length of the detected isoforms correlated with the number of exons (r = 0.43).

A small number of transcripts was qualified as antisense [n = 38; 0.3%], intergenic [n = 109; 0.8%] and genic genomic [n = 6; 0.04%]. All isoform categories were mostly represented by coding transcripts (96–98%), except for the antisense and intergenic categories. Antisense and intergenic transcripts had 60% and 75% predicted ORFs, respectively. Nonsense-mediated mRNA decay (NMD) prediction showed that novel transcripts (27% of the NIC and 21% of the NNC) were more likely to be targeted by the NMD pathway than known transcripts (3% of the FSM).

### 3.4. UTR Extensions

The 5′-end of 46% of the FSM transcripts overlapped completely or almost completely the transcriptional start site (TSS) of the matched reference transcripts (Figure 3A). Similarly, the 3′-end of 51% of the FSM transcripts overlapped completely or almost completely the transcriptional termination site (TTS) of the matched reference transcripts (Figure 3B). Seventeen percent of the FSM and 14% of the ISM transcripts extended beyond the known TSS, while 20% of the FSM and 11% of the ISM transcripts extended beyond the known TTS (Figure 3C,D). The 5′-end of the FSM transcripts were extended by an average of 72 bp (max: 3115 bp). The 3′-end of the FSM transcripts were extended by an average of 505 bp (max: 14,447 bp).

### 3.5. Characterization of Splice Junctions

Novel genes were enriched for multi-exon transcripts, compared to known genes (Appendix A). Of these multi-exonic novel genes, 44.3% (4702) were non-coding. The number of exons per isoform structural class was similar for both known and novel transcripts (Appendix A). Only 623 transcripts were classified as mono-exonic. Their length was comparable to multi-exon transcripts (Appendix A). Almost all splice junctions were canonical (GT-AG, GC-AG or AT-AC dinucleotide pairs) (Table 6). The ten most frequent splice junctions found in the Iso-Seq transcripts are shown in Figure 4A.

Canonical splice junctions from previously annotated genes accounted for 96.89% of all splice junctions (Table 6). New combinations of donor and acceptor splicing sites accounted for 0.7% of the isoforms only. Many of those were found in the novel isoform category, as illustrated in Figure 4B. The top ten splice junctions are shown in Figure 4A.

### 3.6. Novel Isoforms

Novel isoforms comprised 24% of all transcripts. While these 3428 transcripts were classified as novel, 1886 (55%) could still be linked to previously annotated genes because they shared splice junctions with known gene transcripts. These were classified as NIC. The remaining novel transcripts, predominantly classified as NNC, had at least one novel donor or acceptor site (1341 [40.8%] of all novel transcripts]). Intron retention was observed in 34% of the novel isoforms.

The novel transcripts contained 3428 ORFs with a length ≥ 100 amino acids. A total of 105 protein-coding transcripts mapping to 147 novel genes were predicted. These protein-coding ORFs from novel genes had an average length of 327 amino acids. Fifty-four of the predicted proteins matched proteins in the nr database with more than 85% identity and 80% coverage (Appendix A). Appendix A shows the similarity between the novel transcripts identified and the orthologous gene transcripts. These transcripts are absent from the EquCab3.0 NCBI annotation release 103.

### 3.7. Fusion Genes

Forty-eight fusion transcripts were associated with 30 previously annotated genes, most of them originating from read-through transcription of two adjacent genes. The fusion gene *MPDU1_CD68* is displayed in Figure 4B as an example. The majority of fusion transcripts encompassed only two genes. ORFs were predicted in 45 of them. Twenty-six ORFs spanned the coding sequence of the two loci they overlapped. An NMD effect was predicted in 12 fusion transcripts. We also identified 13 fusion transcripts as potential “conjoined genes” based on their sequence matching with the ConjoinG database [20] (Appendix A). Conjoined genes give rise to transcripts combining exons from two or more distinct genes lying on the same strand of a chromosome. The conjoined genes found in our dataset were supported by at least one mRNA or expressed sequence tag (EST) sequence.

### 3.8. Gene Ontology Analysis

The 500 most expressed genes in our dataset were predominantly involved in neutrophil degranulation (GO:0043312), neutrophil activation (GO:0002283) and neutrophil-mediated immunity (GO:0002446). Tissue ontology analysis (Jensen database [27]) showed these genes were overexpressed in trachea, blood and monocytes. Gene ontology analysis using the PanglaoDB database [27] suggested a high expression in macrophages, monocytes, alveolar macrophages and dendritic cells. The enrichment table is detailed in Appendix A.

## 4. Discussion

We used PacBio long-read Iso-Seq RNA-seq to characterize full-length cDNA sequences in equine BALF cells. We were able to identify and map 14,234 transcripts with considerable isoform diversity, of which 4449 were not represented in the current equine genome NCBI annotation release 103. This highlights the need for a more comprehensive and tissue-specific transcriptome annotation in horses.

The coding transcripts identified in our Iso-Seq dataset covered 60% of the 21,129 coding transcripts from the EquCab3.0 NCBI annotation release 103. The levels of gene expression appear to be tissue dependent: using PacBio sequencing in chicken, Kuo et al. [10] identified 211,292 transcripts from the brain but only 14,776 transcripts from the embryo. Hence, our incomplete transcriptome representation most likely reflects a BALF cell-specific gene expression. The results of the gene ontology analyses using tissue (Jensen database) and cell type (PanglaoDB) databases also support this hypothesis.

Our dataset comprised 70% of FSM transcripts, indicating that RNA degradation was minimal. Still, ISM isoforms comprised 9.4% of the Iso-Seq transcripts associated with known genes, similar to the 8.7% reported in the equine FAANG transcriptome [14]. ISM transcripts were supported by reference transcripts with missing 5′- or 3′-exons. These ISM transcripts may represent technical artifacts, for example due to RNA degradation or incorrect priming, or real additional isoforms. These transcripts all had CCS sequences with a median ORF length of 450 bp, indicating they could encode proteins. To which extent these transcripts contribute to the BALF proteome needs to be further explored. A high content of adenosine nucleotides in the RNA strands can lead to mispriming during reverse transcription. However, only 241 of the 1333 ISM reads had 80% or more A nucleotides in the 20 bases flanking their mapped position. Almost all splice sites detected in the ISM transcripts were canonical, making false splice site predictions unlikely. Indeed, splice sites are easily predicted, since 98.7% of the canonical splice sites in mammalian genomes are known to be GT-AG [28]. In summary, ISM transcripts identified with Iso-Seq RNA-seq likely represented biologically true novel transcript isoforms.

The isoform-to-gene ratio of our Iso-Seq dataset was 2.2 when considering protein-coding genes only. This is comparable to the existing NCBI and Ensembl EquCab3.0 reference annotations with a ratio of 2.6 and 2.1, respectively. In contrast, the equine FAANG transcriptome [14] achieved an isoform-to-gene ratio of 4.2 by combining the two existing annotations with an Iso-Seq transcriptome. Peng et al. [14] used an interesting approach where the transcripts only differing at their 5′-ends were collapsed into a single transcript. They then used short-read and ATAC-seq data from the same tissues to assess the completeness of the 5′-end of the isoforms. In the absence of other type of datasets to compare to, we filtered out the transcripts with differing 5′-ends, which may have reduced the number of isoforms recovered.

The GO analysis using the Enrichr database highlighted biological processes that are closely related to the pathophysiology of equine asthma. One of the two horses sampled for this experiment was affected with neutrophilic severe equine asthma. This most likely explains the enrichment of genes associated with neutrophilic inflammation in our dataset. Our Iso-Seq transcripts will thus be valuable when interpreting RNA-seq data originating from asthmatic horses. In future experiments, it would be interesting to extend the panel of transcripts by sampling horses with other forms of asthma, such as mild/moderate mastocytic or eosinophilic equine asthma.

A small proportion (0.3%) of the transcripts were fusion transcripts, most of them due to read-through transcription of two adjacent genes. Peng et al. [14] similarly found them in small numbers (1.21% of all transcripts). The biological significance of fusion transcripts remains unclear. Their hits to the conjoined genes database suggest a functional role, as several of these conjoined genes are conserved among vertebrates [26].

While this study identified many equine BALF cell transcripts not referenced in the current EquCab3.0 NCBI annotation release 103, it also had several limitations. First, the Iso-Seq RNA-seq protocol entails poly(A) selection, resulting in a 3′-bias. Complementary techniques would be required to capture non-polyadenylated RNA such as ribosomal RNA. Consequently, our dataset contains mostly protein coding transcripts and polyadenylated long non-coding RNA transcripts (lncRNAs).

Another potential limitation of our experiment is the sequencing depth. While the sequencing saturation was appropriate for gene detection, it may not have been sufficient to capture the full extent of isoform diversity (see Appendix A). We propose to integrate this equine BALF-specific transcriptome to the recently built equine FAANG transcriptome [14]. In future studies, it will be beneficial to sequence RNA from additional tissues to cover a wider range of cell types and/or developmental stages with potentially different expression profiles [10,29].

Eventually, we acknowledge that the complexity of the different isoforms for some genes is daunting. It is difficult to determine whether this reflects the complexity of equine BALF cell biology, or if it arises from technical artifacts inherent to the Iso-Seq sequencing technology. The replication of long-read sequencing experiments in horses will make it possible to cross-check the data and to disentangle technical artifacts from biological reality. The combination of long-read sequencing with single cell sequencing should enable the generation of high-quality gene expression profiles at a superior resolution.

## 5. Conclusions

In summary, this study demonstrates the potential of long-read sequencing to improve the annotation of the equine transcriptome by providing full-length transcripts at the isoform level. Here, we provide a BALF cell-specific transcriptome that will be useful for future equine bulk or single cell RNA-seq studies.

## Figures and Tables

**Figure 1 genes-13-01722-f001:**
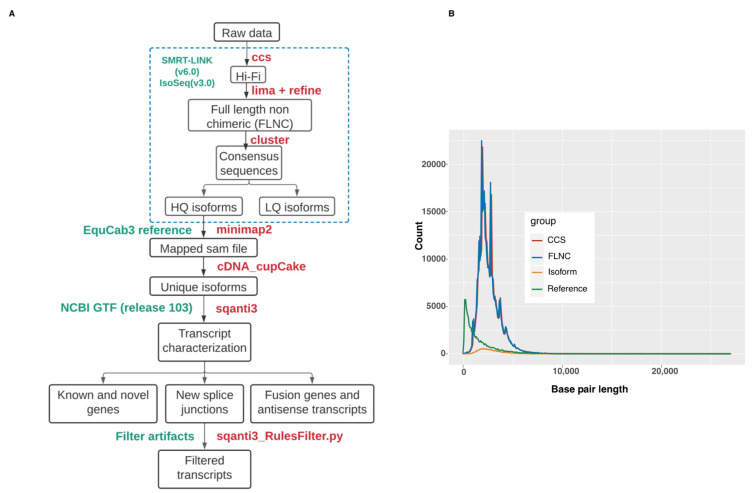
(**A**) Schematic overview of the bioinformatics workflow used to generate full-length transcript annotations. (**B**) Length comparison between reference transcript length and Iso-Seq data.

**Figure 2 genes-13-01722-f002:**
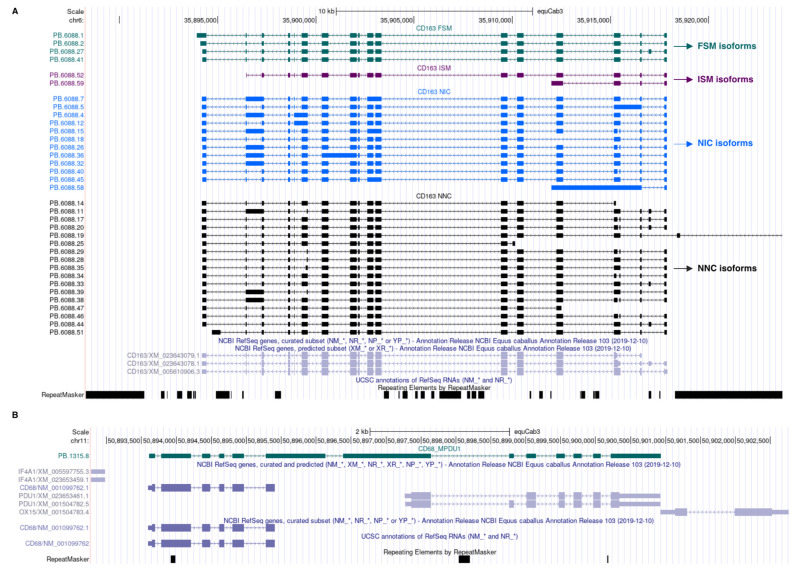
(**A**). UCSC genome browser track of *CD163* in equine BALF cells. (**B**). UCSC genome browser track showing the transcripts of the fusion gene *MPDU1_CD68* (green track). These transcripts cover both the genes *MPDU1* and *CD68*. Their biological significance is unknown at this time.

**Figure 3 genes-13-01722-f003:**
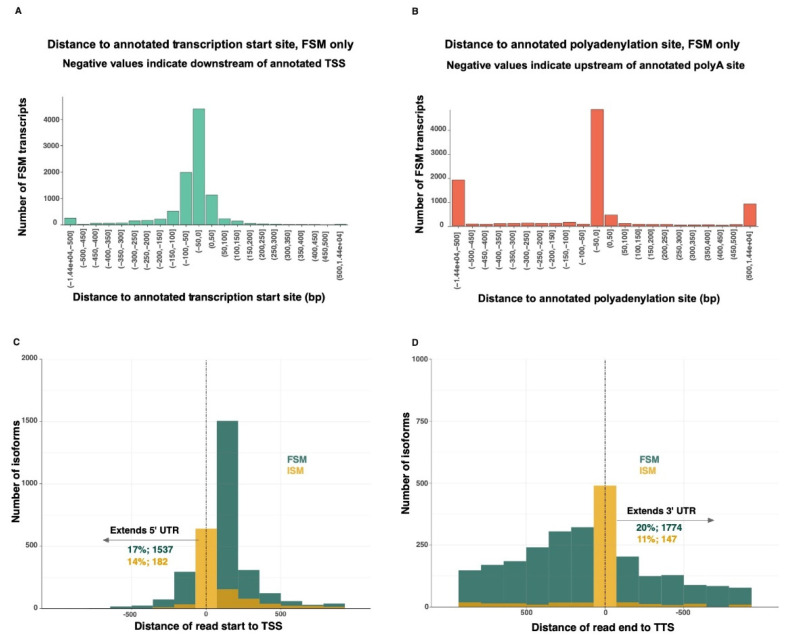
(**A**). Binned distances between the Iso-Seq transcription start sites (TSS) and the corresponding annotated TSS (FSM transcripts only). (**B**). Binned distances between the Iso-Seq transcript poly(A) site and the corresponding annotated poly(A) site (FSM transcripts only). (**C**). In contrast to FSM reads, most ISM reads do not extend beyond the 5′-UTRs of known transcripts. (**D**). FSM and ISM reads extending beyond the 3′-UTRs of known transcripts. TSS: transcriptional start site. TTS: transcriptional termination site.

**Figure 4 genes-13-01722-f004:**
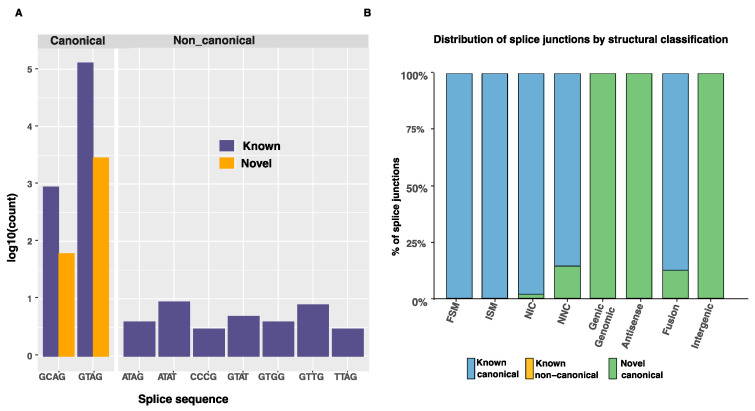
(**A**). Top ten splice sites identified in the known (previously annotated) and novel transcript isoforms. (**B**). Distribution of splice junction types across transcript structural classes.

**Table 1 genes-13-01722-t001:** Characteristics of the horses used for the study.

Sample	1	2
Health status	Asthmatic	Control
Breed	Selle Français	Swiss Warmblood
Sex	Mare	Mare
Age (years)	21	12
HOARSI *	4	1
Weight (kg)	634	613

* The Horse Owner Respiratory Signs Index (HOARSI) was used assess the equine asthma severity. HOARSI 1 indicates a healthy, and HOARSI 4 a severely asthmatic horse.

**Table 2 genes-13-01722-t002:** Transcript structural classes provided by SQANTI3.

Full splice match (FSM)	Isoforms matching perfectly to annotated transcripts
Incomplete splice match (ISM)	Isoforms matching to a subsection of an annotated transcript
Novel in catalog (NIC)	Isoforms with a new combination of annotated splice sites
Novel not in catalog (NNIC)	Isoforms with at least one novel splice site along with annotated splice sites
Intergenic	Isoforms mapping to intergenic region
Genic intron	Isoforms contained within an intron
Genic genomic	Isoforms overlapping with exons and introns
Fusion	Isoforms spanning two annotated loci
Antisense	Isoforms mapping to the complementary strand of an annotated transcript

**Table 3 genes-13-01722-t003:** Summary statistics of PacBio Sequel IIe transcript sequencing data.

Sequence Type	Total Number	Min Length	Average Length	Max Length	N50
Polymerase read	419,674	-	68,058	-	126,248
Subread	12,487,670	500	2244	236,862	2427
CCS	313,563	500	2568	15,078	2759
FLNC	258,022	500	2536	15,000	2689
HQ transcripts	20,462	502	2931	10,513	2937

**Table 4 genes-13-01722-t004:** Overview of the annotated long-read equine BALF cells transcriptome.

Characteristics	Number (Percentage)
Isoforms	14,234
Unique genes	7017
Previously annotated genes	6880 (98.0%)
Novel genes	137 (2.0%)
Genes with >1 isoform	3400 (48.4%)
Genes with >6 isoforms	164 (2.3%)
Known coding transcripts	10,665 (75.0%)
Known non-coding transcripts	141 (1.0%)
Novel coding transcripts	3235 (22.7%)
Novel non-coding transcripts	193 (1.3%)

**Table 5 genes-13-01722-t005:** Structural classes of the 14,234 transcripts identified.

Structural Class	#Isoforms (Percentage)
All	14,234 (100%)
FSM	9473 (66.5%)
ISM	1333 (9.4%)
NIC	1886 (13.2%)
NNC	1341 (9.4%)
Genic genomic	6 (0.04%)
Antisense	38 (0.3%)
Fusion	48 (0.3%)
Intergenic	109 (0.8%)
Genic intron	0 (0%)

**Table 6 genes-13-01722-t006:** Splice junction classification.

Splice Junction Category	Number (Percentage)
Known canonical	67,607 (96.9%)
Known non-canonical	30 (0.04%)
Novel canonical	2171 (3.1%)
Novel non-canonical	0 (0.0%)

## Data Availability

The data presented in this study are available online at: https://www.ebi.ac.uk/ena/browser/view/PRJEB51962; accessed on 14 Februry 2022, (run accession: ERR9954028). The code used for the analysis and the final read annotation can be found at https://github.com/vetsuisse-unibe/EquCab3-Iso-seq-BALF-paper; accessed on 14 Februry 2022.

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
