# Peer review of "Long-Read Transcriptome of Equine Bronchoalveolar Cells"

_genes, 2022, doi:10.3390/genes13101722_

Round 1

Reviewer 1 Report

Good logistics of presenting the issues, methods and research results will undoubtedly contribute to the reader's interest. It is obvious that further research should be aimed at studying transcripts in equine BALF cells, while it is important to take into account the predisposition to asthma and the specifics of the immune response in horses of different specialization.

Author Response

Good logistics of presenting the issues, methods and research results will undoubtedly contribute to the reader's interest. It is obvious that further research should be aimed at studying transcripts in equine BALF cells, while it is important to take into account the predisposition to asthma and the specifics of the immune response in horses of different specialization.

Ans: Thank you for taking the time to review our paper and your appreciative comment. We have also worked on improving language quality throughout the manuscript.

Reviewer 2 Report

In my opinion, although the study by Sage et al. is rather a “technical note” focused on generating better annotated BALF cells transcriptome using PacBio sequencing, the presented results can be valuable – especially for researchers concentrated on the molecular background of equine asthma. However, I have several questions and comments, mainly to the experimental design and “visual” site of the evaluated paper.

1.    Introduction. In my opinion, you should extend this part with information about the importance of functional annotation of animal genomes, including the domestic horse. The equine FAANG consortium has already published several papers (e.g. describing EcuCab 3.0 assembly generation or showing the creation of biobanks of equine tissues for gene expression studies), which definitely corresponds to the primary goal of your study. There is also a nice preprint by Peng et al. showing the utility of long-range sequencing in the better annotation of the equine genome. https://www.biorxiv.org/content/10.1101/2022.06.07.495038v1

I think you should build a better background for your “story” and refer to the aforementioned publications.

2.    Chapter 2.1. Study population. I think this part should contain all information about the studied animals, including their number, breeds, age, health status, etc. Some of these data are presented for the first time in other parts of the manuscript, including Results (e.g., Table 1), which is not logical, in my opinion. All data regarding tested animals should be included in the Material and Methods section.   

3.    Chapter 2.5. How many RNA samples were extracted and assessed before you selected those two for PacBio sequencing? What was the average RIN value and its SD?

4.    I think most of the Discussion section is a bit “descriptive” or “speculative” – which is good. Still, I strongly recommend deeper comparing your results with the studies of other authors. You can also describe shortly the current knowledge regarding the molecular background of equine asthma to show that your study can be an essential step to understanding the biology of the disease.

5.    You should also work on the visual presentation of your paper. For example, you are not consequent with using capital letters in the headings, legends, and Figures/Tables. For example, a legend to Figure 3 (Known, Novel, canonical etc.) or Figure 1A: new splice junctions, Known and Novel genes etc. This technical issue is present also in other parts of the paper. I think the best way is to use a capital letter only for the first word of the given form. But generally, you should be more consequent here. You can also work more on improving language, e.g. line 386 data set à dataset. Line: 90 “Samples were stored for 146 and 342 days for control and asthmatic horse, respectively. Afterwards, cryovials(…) were rapidly thawed(…)”         

6.    Were generated data stored in any public repository? If yes, a corresponding accession number should be provided in the manuscript text.

Author Response

  1. Introduction. In my opinion, you should extend this part with information about the importance of functional annotation of animal genomes, including the domestic horse. The equine FAANG consortium has already published several papers (e.g. describing EcuCab 3.0 assembly generation or showing the creation of biobanks of equine tissues for gene expression studies), which definitely corresponds to the primary goal of your study. There is also a nice preprint by Peng et al. showing the utility of long-range sequencing in the better annotation of the equine genome. https://www.biorxiv.org/content/10.1101/2022.06.07.495038v1

I think you should build a better background for your “story” and refer to the aforementioned publications.

Ans: We thank the reviewer for the comment and pointing us to the preprint. We cited the reference and added the following paragraphs to the introduction section:

“Complete cDNA molecules can now be sequenced using long-read sequencing plat-forms like Pacific Biosciences [7] and Oxford Nanopore [8]. In contrast to short-read sequencing, long-read-based techniques allow unambiguous mapping to a reference annotation, conservation of isoform information, as well as discovery of novel genes, fusion transcripts and non-coding RNA. This has led to significant improvement of the transcriptome annotation in a few domestic species such as chicken [9], rabbits [10] or pigs [11]. The FAANG consortium, whose goal is to create reference functional maps of livestock animal genomes [12] is already taking advantage of long-read sequencing. Recently, Peng et al. [13] applied the PacBio Iso-Seq sequencing technique to nine different horse tissues, including lung. They combined their Iso-Seq transcriptome with the existing NCBI and Ensembl EqCab3.0 reference annotations. The resulting FAANG equine transcriptome comprises 36,239 protein-coding genes and 153,492 transcripts, a considerable improvement compared to the current NCBI annotation, which contains only 21,129 protein-coding genes and 72,102 transcripts. The discovery of novel genes and isoforms with potential biological relevance will benefit to all genomics studies.

Advanced transcriptomics are becoming increasingly popular to study the pathophysiology of equine asthma. Our group recently performed the first single-cell mRNA sequencing of equine BALF [14], a technique that is widely used to study the transcriptome of individual cells and compare them. Peng et al. [13] have also showed that major isoform (based on relative expression values) differs between tissue type, hence a BALF-specific reference transcriptome would be highly desirable.”

  1. Chapter 2.1. Study population. I think this part should contain all information about the studied animals, including their number, breeds, age, health status, etc. Some of these data are presented for the first time in other parts of the manuscript, including Results (e.g., Table 1), which is not logical, in my opinion. All data regarding tested animals should be included in the Material and Methods section.   

      Ans: We have moved the Table 1 and information about studied animals from results (section 3.1) into the study population section 2.1.  

      We elected to present the characteristics of the two horses selected for the experiment only, since the other horses were not sequenced. 

  1. Chapter 2.5. How many RNA samples were extracted and assessed before you selected those two for PacBio sequencing? What was the average RIN value and its SD? 

      Ans: The average RIN was 9.86, with an SD of 0.10. The median was 9.85 with an IQR of 0.13. Since we only had 4 values, we reported the median and IQR (see Methods section 2.5 line 115.

  1. I think most of the Discussion section is a bit “descriptive” or “speculative” – which is good. Still, I strongly recommend deeper comparing your results with the studies of other authors. You can also describe shortly the current knowledge regarding the molecular background of equine asthma to show that your study can be an essential step to understanding the biology of the disease. 

      Ans: The discussion has been reworked according to your recommendations. We elected to compare our results mostly to those of the Peng et al. preprint, as comparison within the same species is most relevant.

      We preferred not to elaborate on equine asthma in the discussion because it seemed too far from the theme of the paper (which is, as you mentioned, pretty technical).

  1. You should also work on the visual presentation of your paper. For example, you are not consequent with using capital letters in the headings, legends, and Figures/Tables. For example, a legend to Figure 3 (Known, Novel, canonical etc.) or Figure 1A: new splice junctions, Known and Novel genes etc. This technical issue is present also in other parts of the paper. I think the best way is to use a capital letter only for the first word of the given form. But generally, you should be more consequent here. You can also work more on improving language, e.g. line 386 data set à dataset. Line: 90 “Samples were stored for 146 and 342 days for control and asthmatic horse, respectively. Afterwards, cryovials(…) were rapidly thawed(…)”       

      We are thankful to the reviewer for noticing these errors. Figures and text were modified accordingly. We worked on improving language quality throughout the manuscript. 

  1. Were generated data stored in any public repository? If yes, a corresponding accession number should be provided in the manuscript text.

      This information can be found in the “Data Availability Statement” line 456:

“The data presented in this study is available online at: https://www.ebi.ac.uk/ena/browser/view/PRJEB51962 (run accession: ERR9954028). The code used for the analysis and the final read annotation can be found at https://github.com/vetsuisse-unibe/EquCab3-Iso-seq-BALF-paper.”

Additional comments: we modified the section 2.3 of the Material and Methods section, as we recently published a study using the same protocol for BALF cryopreservation and thawing (we cited the paper).

Round 2

Reviewer 2 Report

I think the paper is now improved significantly and can be accepted for publication. My last technical suggestion is not to use the word "population" to describe a set of just two horses. I think you can change the title of chapter 2.1 to "Animals"  and consequently correct the title of Table S1.

Good job!

Author Response

I think the paper is now improved significantly and can be accepted for publication. My last technical suggestion is not to use the word "population" to describe a set of just two horses. I think you can change the title of chapter 2.1 to "Animals"  and consequently correct the title of Table S1.

Good job!

Ans: Thank you very much again for your time and your appreciative comment. We have changed the word 'population' to 'animals' in the main text and table S1.